# Using laboratory data to assess the impact of coronavirus (COVID-19) on reflex cryptococcal antigenaemia (CrAg) testing in South Africa

Naseem Cassim[1,2]*, Lindi Marie Coetzee[1,2], Manuel Pedro da Silva[2], Wendy Susan Stevens[1,2], Deborah Kim Glencross[1,2]

1 Wits Diagnostics Innovation Hub (DIH), Faculty of Health Sciences, University of Witwatersrand, Johannesburg, South Africa, 2 National Health Laboratory Service (NHLS), National Priority Programme (NPP), Johannesburg, South Africa

☯ These authors contributed equally to this work.

* naseem.cassim@wits.ac.za

**Data Availability Statement:** The authors do not have permission to share the CrAg data. Data is available from the National Health Laboratory

## Abstract

### Background

Coronavirus disease (COVID-19), caused by the severe acute respiratory syndrome coronavirus 2 (SARS-CoV-2), was first reported in Wuhan, China. Due to the rapid spread globally, it was declared a pandemic in March 2020. Social distancing and lockdown measures were introduced to limit transmission. These strategies could potentially impact the diagnosis and treatment of patients with advanced HIV who are susceptible to secondary infections like cryptococcal disease. In South Africa, reflexed cryptococcal antigenaemia (CrAg) testing and pre-emptive antifungal treatment are recommended preceding antiretroviral therapy initiation for patients with a CD4<100 cells/μl. This study aimed to assess the impact of COVID-19 on CrAg testing in South Africa.

### Methods

Specimen-level data was extracted for individuals ≥15 years from the National Health Laboratory Services repository for calendar years 2018 to 2021. Test volumes and CrAg positivity were assessed at national and provincial levels, by age category and gender. The percentage change in annual and monthly CrAg test volumes for 2020 and 2021 (during lockdown levels) are compared to data reported for 2018. The monthly median CD4 and the percentage of samples with a count <25, 25–50, 51–75 and >75-<100 cells/μl were assessed.

### Results

Specimen data of 11 944 929 CD4 results included 1 306 456 CrAg tests. Annual CD4 and CrAg test volumes declined by 22.4% and 27.8% for 2020 and 2021 respectively (relative to 2018). There were 23 670 CrAg positive outcomes in 2018 compared to 21 399 (-9.6%) and 17 847 (-24.6%) in 2020 and 2021 respectively. A monthly test volume reduction of up to 36.6%, 35.5%, 36.1% and 13.3% was reported for infection waves one to four. CrAg detection increased from 6.3% in 2018 to 7.5% in 2020. More testing was offered to males

Service Corporate Data Warehouse, subject to an application to the Academic Affairs and Research Office. This application can be made using the online system (https://aarms.nhls.ac.za/).

**Funding:** The authors received no specific funding for this work.

(>56%) with a higher detection rate of 8.1% in 2020. Between 81.0% and 81.8% of testing was for patients aged 20 to 49 years. The monthly percentage of specimens <25 cells/µl ranged from 30.2% (June 2019) to 35.3% (August 2020). Overall, the monthly median CD4 ranged from 39 (IQR: 15–70)(August 2020) to 45 (IQR: 19–72)(March 2019) cells/µl. In 2020, the provincial percentage change in CrAg test volumes ranged from 2.9% to -33.7%.

## Conclusion

Our findings confirmed the impact of lockdown measures on both the absolute number of CrAg tests performed and detection (increase in 2020). A smaller impact on the median CD4 was noted. The long-term impact on patient management in immune- compromised individuals needs further investigation.

## Background

In December 2019, the first reports of novel coronavirus disease (COVID-19) originated from the city of Wuhan, China, [1, 2] identified by sequencing of samples from hospitalised patients with pneumonia [2]. Coronaviruses are enveloped ribonucleic acid (RNA) viruses distributed broadly among humans, other mammals and birds [2]. These viruses cause respiratory, enteric, hepatic and neurological diseases [2]. The severe acute respiratory syndrome coronavirus 2 (SARS CoV-2) is responsible for COVID-19 [2]. By the 30th of January 2020, the World Health Organisation (WHO) declared COVID-19 a public health emergency of international concern (PHEIC) [3, 4] and on the 11th of March 2020, the COVID-19 outbreak was escalated to a global pandemic [5].

Subsequently, public health and social measures (PHSM) were introduced to limit COVID-19 transmission and reduce mortality [6]. This included the following interventions: (i) personal protective measures (hand hygiene, respiratory etiquette and mask-wearing), (ii) environmental measures (cleaning, disinfection and adequate ventilation), (iii) surveillance and response measures (testing, genetic sequencing, contact tracing, isolation and quarantine), (iv) physical distancing measures (regulating the number and flow of people attending gatherings, maintaining distance in public or workplaces and domestic movement restrictions) and (v) international travel restrictions [6]. As a result, many countries introduced social distancing and lockdown rules to control the spread of COVID-19 [7].

On the 5th of March 2020, the first confirmed COVID-19 case was diagnosed in South Africa [8]. When 402 cases had been identified after 18 days, the national government of South Africa announced a national lockdown [9]. to manage the COVID-19 pandemic and decrease the rate of the outbreak [7, 10]. South Africa implemented lockdown levels (ranging from five to one) from 26 March 2020 until 4 April 2022, with varying stringency of movement and travel restrictions [10]. During more stringent levels of lockdown (level 5), international and domestic travel was banned, primary and secondary schools were closed, gathering restrictions were introduced and people were advised to work from home with travel restrictions limited to essential services; social distancing and hand hygiene were made mandatory [9].

The COVID-19 lockdown strategy, social distancing rules and community containment measures are likely to have negatively impacted the diagnoses and treatment of communicable diseases such as HIV, tuberculosis and malaria [7]. A study from South Africa reported that the COVID-19 pandemic may have resulted in challenges such as the diversion of the health workforce, suspension of services, reduced health-seeking behaviour, unavailability of supplies

and the deterioration in data monitoring and funding [11]. Due to preferential treatment of COVID-19 patients and increased funding for the pandemic, the available healthcare resources for HIV care became increasingly limited [7, 12]. Madhi *et al* reported a 22% reduction in the average weekly HIV viral load testing during the level 5 lockdown compared to pre-lockdown periods [13]. Similarly, a 33% reduction in CD4 testing was reported [13]. Another local study reported that for July 2020 (wave one of COVID-19 infections), CD4 testing was 23.9% lower than reported for the same period the previous year, in July 2019 [15]. During wave two of infections, CD4 testing was 21.5% lower in January 2022 compared to the same month in 2019 [14]. These findings confirmed that COVID-19 had a great impact on the diagnosis and treatment of communicable diseases in South Africa, mainly due to restricting access to care during lockdown levels [14].

Cryptococcal meningitis (CM) is a leading cause of death among people living with HIV (PLHIV) that are immunocompromised. PLHIV that have undiagnosed cryptococcal disease at antiretroviral treatment (ART) initiation have a higher mortality risk [15]. Cryptococcal antigenemia (CrAg) testing has demonstrated very good sensitivity and specificity to diagnose cryptococcal meningitis weeks before onset [16, 17]. Two screening strategies are recommended for CrAg, i.e. reflex testing on remnant blood or provider-initiated screening when the patients return for care after CD4 testing [18]. Larson *et al* reported that reflex CrAg screening is likely to be cost-saving or have lower additional costs per additional year of life saved [18]. Therefore, local guidelines recommended the implementation of reflexed CrAg testing for all PLHIV in South Africa with a confirmed laboratory CD4 count of <100 cells/μL [19]. Reflexed CrAg testing is performed on remnant CD4 blood samples with a CD4 100 cells/μL through 47 testing laboratories of the National Health Laboratory Service (NHLS) [18]. Provider-initiated CrAg testing is done on request by a clinician/ health care worker and done through microbiology laboratories of the NHLS [18]. Those with evidence of Cryptococcal infection will receive antifungal treatment before antiretroviral therapy initiation [19].

The NHLS is the largest diagnostic pathology service in South Africa supporting the national and provincial health departments [20–22]. It provides laboratory services to over 80% of the population and performs the majority of HIV, TB and cervical cancer testing through a network of 268 laboratories [20–22].

## Objectives

The primary objective of the study was to assess the impact of COVID-19 on cryptococcal disease test volumes and burden (positivity) at national and provincial levels. Secondary objectives set out to establish the impact of COVD-19 on CrAg test demographics (gender and age) and the median CD4 count.

## Methods

### Ethical considerations

Ethical clearance was obtained from the Human Research Ethics Committee (HREC)(Medical) at the University of the Witwatersrand (M220163). Specimen-level data was extracted without any patient identifiers. Patient consent was not required as secondary laboratory data was used.

### Study design

The cross-sectional study design was used to analyse secondary reflexed CrAg laboratory data, with a CD4 count <100 cells/μl, for the period 1 January 2018 to 31 December 2021.

## Data preparation

Episode data extraction was provided by the NHLS laboratory data repository and included the following variables: (i) episode number, (ii) reviewed date, (iii) age (in years), (iv) gender, (v) province, (vi) CD4 count and (vii) reflexed CrAg result. All rejected and un-reviewed specimens were excluded. The province was populated and managed by the data repository based on the requesting health facility location. The sample reviewed date is electronically generated by the laboratory information system (LIS) when CD4 and CrAg results are verified and released. The age and gender are provided by the health care worker on the laboratory request form and captured on the LIS when specimens are received for testing. The data reported in this paper was not deduplicated due to the absence of a unique patient identifier. Hence, multiple results per patient may be included. An earlier study using de-duplicated patient data reported no significant differences in CrAg positivity, indicating that specimen-level data was sufficient to describe the impact of COVID-19 on episode volumes [23].

The episode data was uploaded in SAS 9.4 (Cary, NC, USA) for data preparation and analysis. The sample reviewed date was used to extract the year and month for each episode number. The year and month were combined to create a new period variable, e.g., 2018|1 for January 2018. The age parameter was categorised as: (i) 15–19, (ii) 20–29, (iii) 30–39, (iv) 40–49, (v) 50–59, (vi) 60–69, (vii) ≥70 and (viii) Unknown (where no age was provided). CrAg data is reported only for episodes with a confirmed CD4 <100cells/μl. The CD4 counts <100cells/μl. were further categorised as <25, 25–50, 51–75 and >75-<100 cells/μl. CrAg results were reported as positive or negative. The detection rate in the paper is a synonym for CrAg positivity/burden and the terms are used interchangeably. Aggregate CD4 test volumes irrespective of count were extracted by calendar year and month.

## Statistical analysis

National CD4 and CrAg test volumes were reported per calendar year and month, with the CrAg positivity calculated nationally and per province. Where a province variable was not captured in the data repository, it was excluded from analysis. For CrAg-positive samples, the distribution of tests by gender and age categories were analysed. The chi-squared test was used to assess the association between the CrAg result and calendar year, age category and gender, assuming an alpha of 0.05. The monthly test volumes for 2020 and 2021, when lockdown levels were in place, were compared to the corresponding month in 2018 (pre-COVID-19); the calculated percentage change is represented as a line chart, e.g., percentage change between January 2020 and 2018.

The number of SARS CoV-2 cases was reported as a bar chart [24]. This data was used to assign outbreak waves based on the National Institute for Communicable Diseases (NICD) definition (COVID-19 weekly incidence ≥ 30 cases per 100 000 persons) as follows: (i) one: June to August 2020, (ii) two: November 2020 to February 2021, (iii) three: May to September 2021 and (iv) four: December 2021 [25]. Data for the months of 2022 during wave four are out of the scope of this study [23].

For 2020 and 2021 when lockdown measures were introduced, we determined the percentage change in CrAg test volumes by year and month. The highest monthly percentage change in each wave was determined at the national level. We assessed whether the monthly median CD4 (for a count <100 cells/μl) and CrAg positivity differed for two periods defined as pre-(January 2018 to February 2020) and COVID-19 (March 2020 to December 2021). The monthly percentage of specimens with a count <25, 25–50, 51–75 and >75-<100 cells/μl was calculated per testing year. Annual CD4 test volumes, irrespective of the count, were assessed to determine the percentage change between 2018 and 2021 and compared to the percentage

**Table 1. Annual CD4 and CrAg test volumes, percentage contribution, positivity rate and percentage change (relative to 2018) is reported.**

| Year | CD4 Volumes | % Change | CrAg Volumes | % Change | % CrAg of Total CD4[#] | CrAg Positive Volumes (%) | p-value |
|------|-------------|----------|--------------|----------|------------------------|---------------------------|---------|
| 2018 | 3 357 344 (28.11) | | 378 359 (28.96) | | 11.27 | 23 670 (6.26) | |
| 2019 | 3 247 028 (27.18) | -3.3% | 361 553 (27.67) | -4.4% | 11.13 | 23 134 (6.40) | |
| 2020 | 2 767 086 (23.17) | -17.6% | 293 477 (22.46) | -22.4% | 10.61 | 21 399 (7.29) | |
| 2021 | 2 573 471 (21.54) | -23.3% | 273 067 (20.90) | -27.8% | 10.61 | 17 847 (6.54) | |
| **Total** | **11 944 929 (100)** | | **1 306 456 (100)** | | **10.91** | **86 050 (6.62)** | **≤0.001** |

[#]Percentage CrAg tests of total CD4 volumes per year

CrAg: Cryptococcal antigen

Source: Authors own work

change in reflexed CrAg testing for a CD4 count <100 cells/μl. The contribution of reflexed CrAg to CD4 test volumes was also assessed.

# Results

## National results

Over the four-year test period, a total of 11.9 million CD4 tests were performed across 47 NHLS laboratories. The percentage contribution per testing year of the total changed from 28.1% in 2018 to 21.5% in 2021 (Table 1). The number of CrAg tests performed (1.3 million over 4 years) made up between 10.6 and 11.3% of the total CD4 test volumes per annum, with a national average of 10.9%. CrAg positivity ranged from 6.3 to 7.3% with a national average of 6.6% for the test period. A chi-squared test for the association between the test year and CrAg result, reported a significant difference (p≤0.001). The percentage change in CD4 test volumes ranged from -3.3% (2019) to -23.3% (2021) and -4.4% to -27.8% for CrAg.

## Demographics analysis of positive CrAg results

Analysis of the number of CrAg positive tests per annum, reported no substantial changes in gender distribution, with males making up 56% of tests performed (Table 2). An unknown

**Table 2. Gender and age category distribution for positive reflexed CrAg samples, with the positivity rates also indicated.**

| | CrAg Positive samples n = (%) | | | | CrAg Positivity | | | |
|--|------|------|------|------|------|------|------|------|
| | 2018 | 2019 | 2020 | 2021 | 2018 | 2019 | 2020 | 2021 |
| **Gender** | | | | | | | | |
| Female | 9 991 (42.2)) | 9 776 (42.3) | 9 070 (42.4) | 7 434 (41.7) | 5.5% | 5.6% | 6.4% | 5.8% |
| Male | 13 307 (56.2) | 13 002 (56.2) | 12 021 (56.2) | 10 152 (56.9) | 7.0% | 7.1% | 8.1% | 7.2% |
| Unknown | 372 (1.6) | 356 (1.5) | 308 (1.5) | 261 (1.5) | 6.9% | 6.3% | 7.2% | 6.6% |
| **Age Category** | | | | | | | | |
| 15–19 | 502 (2.1) | 514 (2.2) | 467 (2.2) | 369 (2.1) | 5.4% | 5.7% | 6.6% | 5.9% |
| 20–29 | 3 790 (16.0) | 3 377 (14.6) | 3 103 (14.5) | 2 642 (14.8) | 5.9% | 5.7% | 6.5% | 6.3% |
| 30–39 | 9 214 (38.9) | 9 093 (39.3) | 8 422 (39.4) | 6 619 (37.1) | 6.5% | 6.6% | 7.5% | 6.6% |
| 40–49 | 6 276 (26.5) | 6 270 (27.1) | 5 983 (28.0) | 5 192 (29.1) | 6.4% | 6.7% | 7.9% | 7.2% |
| 50–59 | 2 204 (9.3) | 2 305 (10.0) | 2 104 (9.8) | 1 856 (10.4) | 5.7% | 6.1% | 6.8% | 5.9% |
| 60–69 | 592 (2.5) | 612 (2.6) | 556 (2.6) | 518 (2.9) | 5.0% | 5.1% | 5.9% | 5.0% |
| ≥70 | 92 (0.4) | 83 (0.4) | 124 (0.6) | 93 (0.5) | 4.2% | 3.7% | 6.5% | 3.7% |
| Unknown | 1 000 (4.2) | 880 (3.8) | 640 (3.0) | 558 (3.1) | 7.5% | 7.5% | 7.0% | 7.2% |

Source: Authors own work

CrAg: Cryptococcal antigenaemia

gender contributed <2% of reported results. CrAg positivity for both females and males spiked in 2020 at 6.4 and 8.1% respectively (also mimicked in the Unknown gender group). A p-value of ≤0.001 was reported for the chi-squared test for the association of gender with CrAg results (data not shown).

The CrAg positive distribution per age category only showed year-on-year different for those aged 40 to 49 years. CrAg positivity increased from 2018 to 2019, peaking for all age groups in 2020. By 2021, the CrAg positivity for the majority of age groups returned to 2018/19 values except for the 20–29 and 40–49 age groups. A p-value of ≤0.001 was reported for the chi-squared analysis for the association of age category and CrAg results (data not shown).

## Comparative results before and during COVID-19

**Percentage change in CrAg test volumes.** The monthly percentage change in total CrAg test volumes relative to the respective month in 2018 ranged from -8.7% (February 2020) to -36.6% (August 2020) (Fig 1). For 2021, the monthly percentage change in CrAg test volumes ranged from -13.3% (December 2021) to -36.1% (August 2021). During the first COVID-19 wave, a monthly percentage change of between -29.1% and -36.6% was reported. For the

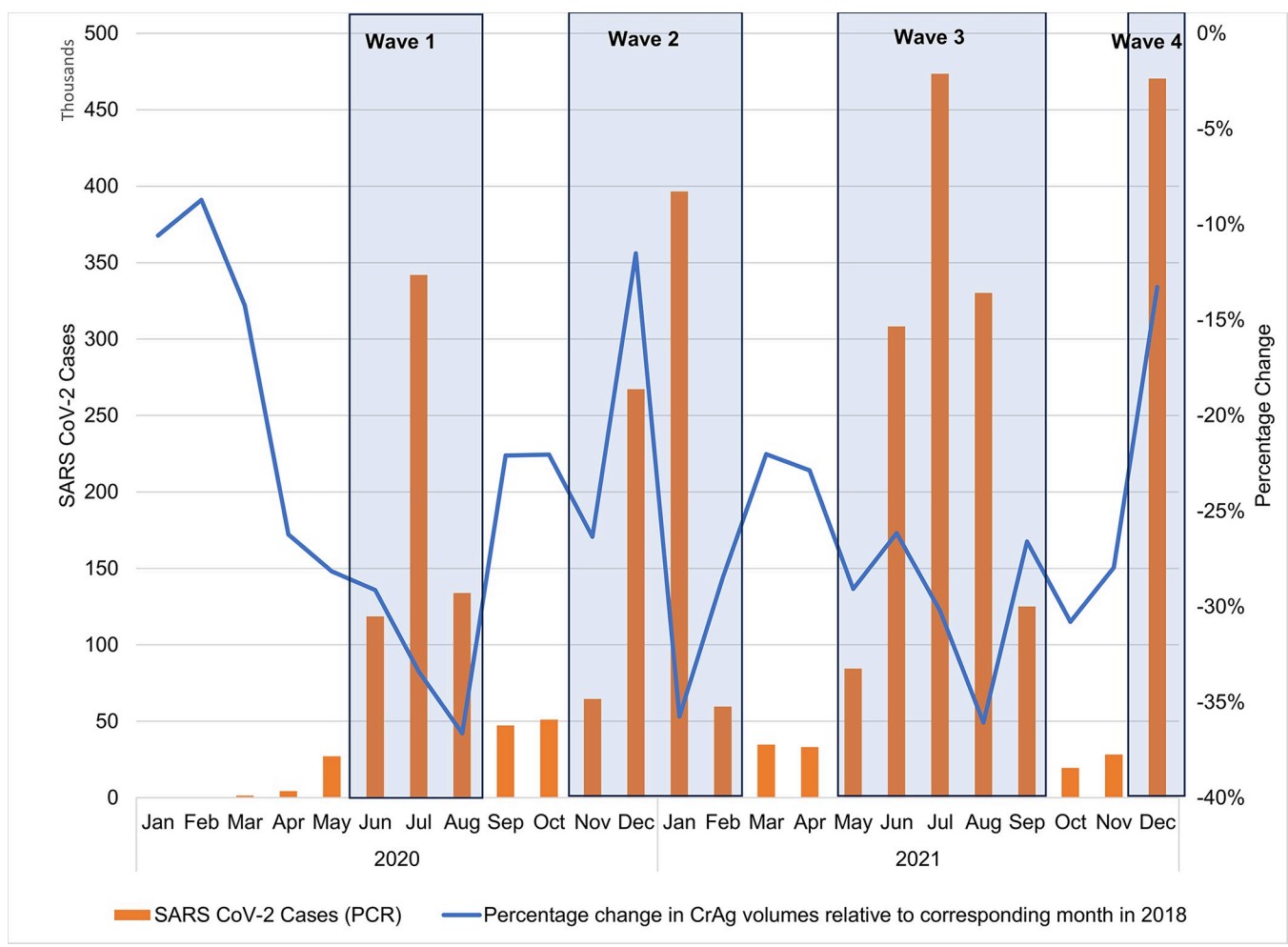

**Fig 1. The monthly percentage change in reflexed CrAg test volumes for 2020 and 2021 were compared to 2018 data.** The number of SARS CoV-2 cases is reported as bar charts, with waves indicated.

second wave, a percentage change of between -11.5% and -35.8% was reported. A percentage change of between -26.2% and -36.1% was reported for wave three, with a percentage change of -13.3% for December 2021 (wave four).

For lockdown levels 5, 4, 3,'2 and 1 in 2020, a percentage change of -25.1%, -25.6%, -31.3%, -31.0%, -22.3% was reported compared to test volumes for 2018 (data not shown) [26].

**Monthly median CD4 analysis (<100 cells/µl).**    Overall analysis of the 4-year test period, reported a median CD4 of 43 (IQR: 18–71), 43 (IQR: 18–71), 43 (IQR: 18–71) and 44 (IQR: 19–72) cells/µl respectively per year (2018 to 2021) (data not shown). Nationally, the monthly median CD4 ranged from 39 (IQR: 15–70)(August 2020) to 45 (IQR: 19–72)(March 2019) cells/µl. For 2018, a monthly median CD4 range of 41 (IQR: 17–69)(December) to 44 (IQR: 19–72)(multiple months) cells/µl was reported (Fig 2). Similarly, the monthly median CD4 ranged from 42 (IQR: 18–71)(January) to 45 (IQR: 19–72)(March) cells/µl for 2019. For 2020 when COVID-19 cases and lockdown measures were introduced, a monthly median CD4 range of 39 (IQR: 15–70) to 45 (IQR: 19–70) was reported. Similarly, for 2021 the monthly median CD4 ranges from 43 (IQR: 19–71) to 45 (IQR: 19–72) cells/µl. When CrAg positivity peaked at 8.0% in April 2020, the median CD4 was 42 cells/µl. The lowest monthly median CD4 reported was 39 cells/µl (IQR: 15–70) in August 2020 with a CrAg positivity of 7.2%. The

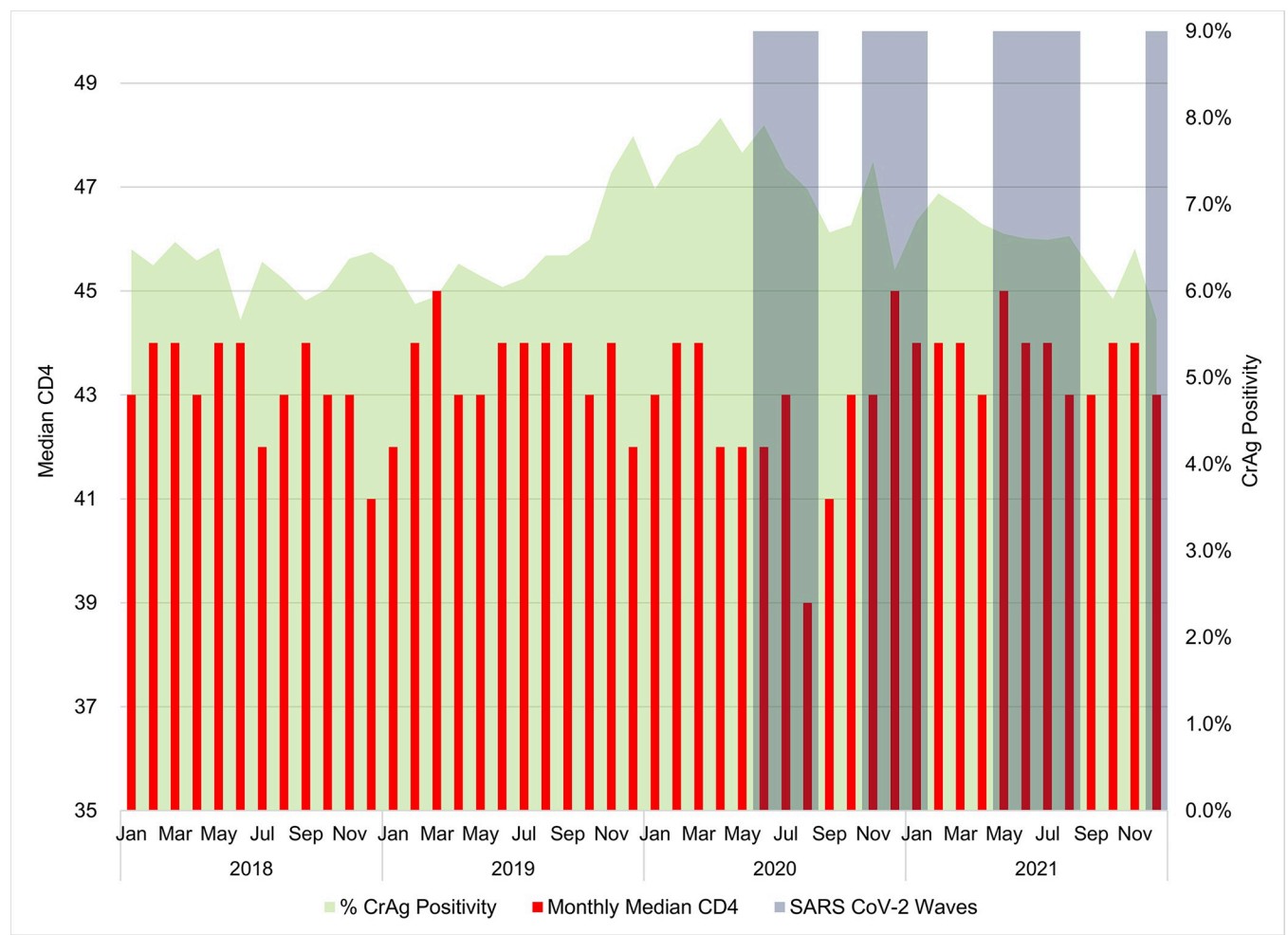

**Fig 2. Monthly median CD4 values for a count <100 cells/µl for 2018 to 2021 are plotted against the reflexed CrAg positivity.**

monthly median CD4 reported a skewness of -1.330031, hence the non-parametric Mann-Whitney test was used (data reported for 48 observations with the exact option used), The results indicated no statistically significant difference between the monthly median CD4 for the two groups (pre-COVID-19 and COVID-19) with a p-value of 0.7431. However, a significant difference was noted for CrAg positivity with a p-value of 0.021.

**Monthly CD4 category analysis (<100 cells/μl).** Overall, there were 31.7%, 24.9%, 22.1% and 21.3% of samples with a CD4 category of <25, 25–50, 51–75 and >75-<100 cells/μl, respectively (Fig 3). The monthly analysis revealed that the percentage of samples <25 cells/μl ranged from 30.2% (June 2019) to 35.3% (August 2020). Similarly, samples with CD4 category of 25–50 cells/μl ranged from 24.0% (February 2021) to 28.1% (December 2020). For the 51–75 and >75-<100 cells/μl categories, a range of 20.3–24.3% and 19.3–22.4% was reported. For August and September 2020, the percentage of samples with a CD4 <25 cells/μl peaked at 35.3% and 33.9% respectively. The CrAg positivity was 7.2% and 6.7% for the same months.

**Provincial analysis.** The province test parameter was not provided for 3 046 samples (0.23%). The comparative percentage change in CrAg test volumes for 2020 versus 2018 at the provincial level ranged from 2.9% (Western Cape) to -33.7% (Free State). Five provinces

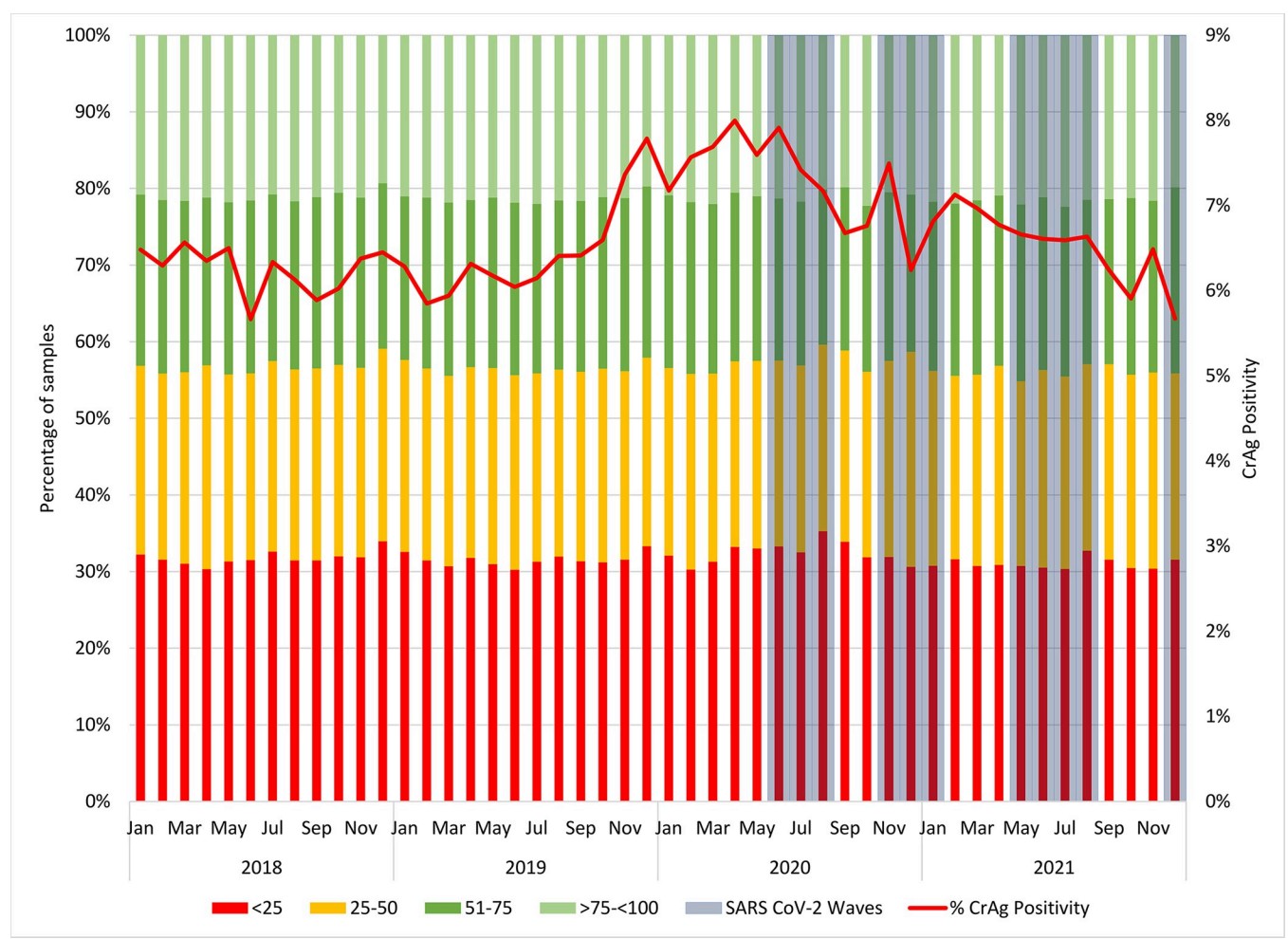

**Fig 3. Monthly percentage tests with a CD4 count <25, 25–50, 51–75 and >75-<100 cells/μl are reported for 2018 to 2021.** The CrAg positivity is also indicated.

**Table 3. Provincial annual CrAg test volumes, positivity and percentage change for calendar years 2020 and 2021 are compared to the correlating data from 2018.**

| Province | 2018 | | 2020 | | | 2021 | | |
|---|---|---|---|---|---|---|---|---|
| | Test Volumes n = | CrAg positivity (%) | Test Volumes n = | % Change* | CrAg positivity (%) | Test Volumes n = | % Change* | CrAg positivity (%) |
| Eastern Cape | 43 612 | 7.3% | 34 613 | -20.6% | 8.2% | 36 225 | -16.9% | 7.9% |
| Free State | 31 518 | 4.3% | 20 893 | -33.7% | 5.8% | 20 919 | -33.6% | 4.9% |
| Gauteng | 133 291 | 5.9% | 94 478 | -29.1% | 6.1% | 80 645 | -39.5% | 5.4% |
| KwaZulu-Natal | 61 958 | 8.2% | 47 499 | -23.3% | 9.7% | 41 611 | -32.8% | 8.8% |
| Limpopo | 27 501 | 5.8% | 23 579 | -14.3% | 7.9% | 20 941 | -23.9% | 6.5% |
| Mpumalanga | 26 793 | 6.4% | 20 953 | -21.8% | 7.0% | 18 137 | -32.3% | 6.2% |
| North West | 21 313 | 4.7% | 18 347 | -13.9% | 5.8% | 17 638 | -17.2% | 5.4% |
| Northern Cape | 6 934 | 3.0% | 6 520 | -6.0% | 3.7% | 6 773 | -2.3% | 3.0% |
| Western Cape | 25 089 | 6.3% | 25 808 | 2.9% | 8.7% | 28 890 | 15.2% | 7.8% |

*Percentage change calculated using test volumes for 2018

Source: Authors own work

CrAg: Cryptococcal antigenaemia

reported a percentage change in CrAg test volumes of ≤-20% (Table 3). The comparison of 2021 and 2018 CrAg volumes reported a percentage change range of 15.2% (Western Cape) and -35.9% (Gauteng), with five provinces reporting a percentage change of ≤-20%.

In 2018, CrAg positivity ranged from 3.0% (Northern Cape) to 8.2% (KwaZulu-Natal). For 2020, this varied from 3.7% (Northern Cape) to 9.7% (KwaZulu-Natal). Similarly, CrAg positivity ranged from 3.0% (Northern Cape) to 8.8% (KwaZulu-Natal) for 2021 (Table 3). Between 2018 and 2020, the provincial CrAg positivity change ranged from 0.2% (Gauteng increased from 5.9% to 6.1%) to 2.4% (Western Cape; increased from 6.3% to 8.7%). Comparing CrAg positivity between 2018 and 2021, the provincial changes varied from -0.5% (Gauteng) to 1.5% (Western Cape).

## Discussion

This study aimed to assess the impact of COVID-19 on cryptococcal disease in South Africa. Aggregate CD4 test volumes showed a significant reduction in 2020 and 2021 compared to 2018. Overall, a declining trend in testing of around 5% per annum was noted pre-COVID-19 due to revised HIV guidelines of "test and treat" introduced in 2016 [19, 27, 28]. However, the impact of COVID-19 exacerbated the test volume reductions for 2020 and 2021. Of greater concern was that the impact on reflexed CrAg testing was even more substantial. The proportion of samples with a count <100 cells/μl did not change substantially over the study period, indicating that the reduction in CrAg test volumes followed trends in CD4 tests.

National CrAg positivity increased by one percent in 2020. Historical data reported a detection range of 4.2% to 5.6% between 2015 and 2017. This highlights that the CrAg positivity in 2020 is incongruent with the pre-COVID-19 annual trends. By 2021, the national CrAg positivity reduced to levels reported before 2020, despite lower CrAg volumes and less restrictive levels imposed, confirming that higher lockdown levels imposed in 2020 resulted in higher detection rates.

While reflex CrAg test volumes reduced substantially, there were no significant demographic changes noted (age and gender). Across the study period, more males received a CrAg result, most likely related to well-described late presentation of males to clinics locally [29–31]. Similarly, the majority of testing was for ages 20 to 49 which is similar to routine CD4 testing.

Therefore, it does not appear that the lockdown restrictions impacted the age and gender composition of those seeking healthcare where CD4 testing was requested.

The monthly analysis for reflexed CrAg testing (<100 cells/μl) showed that during COVID-19 wave one to four, a reduction of up to -36.6% was reported compared to 2018 data, confirming that CrAg volumes were substantially affected by COVID-19 outbreaks.

As CrAg rates of detection peaked in 2020 during the first wave of infections, the median CD4 for a count <100 cells/μl reduced highlighting that patients were more immune-suppressed. i.e. presented later. Subsequently, the median CD4 recovered to levels reported before 2020. In contrast, the distribution of CD4 categories was relatively unaffected by the pandemic. Later presentation indicates that severely ill patients were the ones presenting for care.

Local studies have shown that the percentage of samples with a CD4<100cell/μl and viral suppression (<400 copies/ml) varied per province [29, 32]. NHLS annual reports from 2018–2022 confirmed the range of provincial CD4%<100 cells/μl between 3.1–27% with little variability between years [33–36]. The annual percentage CD4 samples reported with a count <100cells/μl was stable at ~10% with a positivity of 6–7% before 2018 [33–36]. Given the disparity in both very advanced HIV disease and viral suppression previously reported, we assessed provincial CrAg test volumes and positivity for each calendar year of the test period.

The provincial analysis revealed that CrAg test volumes decreased in 2020 and 2021 except for the Northern and Western Cape provinces. In particular, provinces such as the Free State, Gauteng, KwaZulu-Natal and Mpumalanga reported substantial reductions for 2020 exceeding twenty percent. However, for 2021, further reductions of over thirty percent were noted for four provinces, while this trend was only noted in one province in 2020. The provincial variability indicates the varied impact of lockdown in different geographical areas. These findings indicate that lockdown restrictions may have resulted in reduced access as many health facilities closed, especially during levels four and five [11, 12]. The Western Cape was the only province where an increase was noted in CrAg test volumes for 2020 and 2021 relative to previous years. It is not possible to determine the reason for this from the data presented here as it may be multi-factorial. One possibility could be the lower HIV prevalence reported for the Western Cape Province [37].

Overall, the highest CrAg detection rate was reported for the KwaZulu-Natal province, confirmed in earlier studies, even though they reported the lowest percentage CrAg tests of total CD4 tests conducted [23, 38]. However, between 2018 and 2020 the detection rate increased in all nine provinces, which is out of the norm of reported historical CrAg positivity data showing stability over time at provincial level. Additional analysis of CrAg positivity recovery rates at district, sub-district and facility levels is needed to understand the changes reported during COVID-19, particularly in 2020.

It has been argued that strict lockdown measures implemented to curb the spread of SARS_COV-2 infections could have long-lasting public health ramifications that transcend the immediate health risks posed by COVID-19 [11, 12, 39]. The study findings suggest that patients may have delayed seeking HIV care due to several factors including the COVID-19 crisis, overloaded hospitals, limited access, and fear of exposure [39]. The lockdown also resulted in job losses for formal and informal workers [39]. A local study reported that the South African growth domestic product (GDP) was expected to reduce by between 5 and 10 percent during 2020, due to significant job losses reported [39–41]. In times of crisis and loss of income, communities are more likely to prioritise food, water, safety and shelter rather than traveling to a health facility, and this may be indicative of why these patients with very advanced HIV disease did not present for care sooner [13]. This supports the notion that a sustained lockdown has had a prolonged, longer-term negative impact on patients accessing care and receiving testing. Similar findings in respect of restricted access to care and reduced

testing for tuberculosis (TB) have been reported [13]. These findings are in line with WHO reporting, that for the first time, reported a TB-related mortality increase in 2020 [42].

## Conclusion

In conclusion, our findings revealed that there was a negative impact of COVID-19 on the absolute numbers of CrAg tests performed, as well as an increase in detection rate in certain provinces, probably reflecting only severely ill patients with very low CD4 counts, presenting themselves for care. The work highlights the missed opportunity for screening patients with advanced HIV for Cryptococcal disease during the COVID-19 pandemic in South Africa. This work provides evidence of the usefulness of secondary laboratory data to investigate the impact of pandemics on national laboratory testing services.

## Limitations

One of the limitations of this study is that only reported specimen-level data for reflexed CrAg testing is reported. By de-duplicating the data, the CrAg positivity would be either slightly higher or lower. However, the former is unlikely to have significantly influenced the outcomes reported here; an analysis of data for KwaZulu-Natal that performed de-duplication noted very minor differences between positivity and prevalence [23]. A further limitation of this study is that there was no linking of the CrAg and COVID-19 results to develop a cohort due to the absence of a unique patient identifier. This cohort analysis would have generated very valuable insights given the differing incubation periods for COVID-19 and cryptococcosis. Another limitation of this study was that no linking of the CrAg and COVID-19 results was undertaken to assess co-infection and the possible impact that COVID-19 infection would have played in the reduction of CD4 count and further immunosuppression that would have led to a higher likelihood of contracting Cryptococcal disease. Data is only reported for reflex CrAg testing, which is in line with local guidelines [19]. The omission of provider-initiated CrAg testing may introduce some bias. However, a local study reported very high coverage of 95% for reflex screening which indicates that any potential bias by omitting provider-initiated data may be very limited [43]. Another limitation was that data was only extracted for testing for those aged 15 years and above. The decision to exclude the younger ages was based on the weekly surveillance data generated by NICD that reported that the cumulative number of COVID-19 cases from 3 March 2020 to 1 January 2022 was 93% for 15 years and older [44]. Additional analysis may be considered to include those aged 10 to 14 years.

## Supporting information

**S1 File.**
(PDF)

## Acknowledgments

The authors would like to thank all CD4 laboratory staff for providing CrAg testing.

## Author Contributions

**Conceptualization:** Naseem Cassim.

**Data curation:** Naseem Cassim.

**Formal analysis:** Naseem Cassim.

**Investigation:** Naseem Cassim, Lindi Marie Coetzee.

**Methodology:** Naseem Cassim, Lindi Marie Coetzee.

**Supervision:** Wendy Susan Stevens, Deborah Kim Glencross.

**Validation:** Lindi Marie Coetzee.

**Writing – original draft:** Naseem Cassim.

**Writing – review & editing:** Naseem Cassim, Lindi Marie Coetzee, Manuel Pedro da Silva, Wendy Susan Stevens, Deborah Kim Glencross.

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
