## [Decision Letter · Decision Letter 0]

9 Mar 2023

PONE-D-22-31762Using laboratory data to assess the impact of Coronavirus (COVID-19) on reflex cryptococcal antigenaemia (CrAg) testing in South AfricaPLOS ONE

Dear Dr. Naseem Cassim

Thank you for submitting your manuscript to PLOS ONE. After careful consideration, we feel that it has merit but does not fully meet PLOS ONE’s publication criteria as it currently stands. Therefore, we invite you to submit a revised version of the manuscript that addresses the points raised during the review process.

Please submit your revised manuscript by the Apr 23 2023 11:59PM.  If you will need more time than this to complete your revisions, please reply to this message or contact the journal office at plosone@plos.org. Please include the following items when submitting your revised manuscript:A rebuttal letter that responds to each point raised by the academic editor and reviewer(s). You should upload this letter as a separate file labeled 'Response to Reviewers'.A marked-up copy of your manuscript that highlights changes made to the original version. You should upload this as a separate file labeled 'Revised Manuscript with Track Changes'.An unmarked version of your revised paper without tracked changes. You should upload this as a separate file labeled 'Manuscript'.Copies of all gatekeeper and ethics review board approvals.  

We look forward to receiving your revised manuscript.

Kind regards,

Mergan Naidoo, PhD

Academic Editor

PLOS ONE

Journal Requirements:

The authors declare that they have no financial or personal relationships that may have inappropriately influenced them in writing this article.

Reviewers' comments:

Reviewer's Responses to Questions

**Comments to the Author**

1. Is the manuscript technically sound, and do the data support the conclusions?

Reviewer #1: Partly

2. Has the statistical analysis been performed appropriately and rigorously? 

Reviewer #1: No

3. Have the authors made all data underlying the findings in their manuscript fully available?

Reviewer #1: No

4. Is the manuscript presented in an intelligible fashion and written in standard English?

Reviewer #1: Yes

5. Review Comments to the Author

Reviewer #1: On the whole, a well-written article in terms of structure and content.

The research question seems to be along the lines of "What was the impact of COVID-19 on CrAg testing in South Africa?".

Only three issues were detected on language review:

1. "one" instead of "on" p. 4

2. "curve" instead of "curb" p. 13

3. It should be made clear that "CrAg positivity" and "CrAg detection" are synonymous (if I understand correctly).

47 vs. 49 labs? Is it the case that there are 49 CD4 labs, only 47 of which do reflex CrAg testing?

Under "Methods: Statistical Analysis" monthly test volumes are stated to be compared to 2019. In "Results: Percentage change..." it is stated to be compared to the respective month in 2018 - this is confirmed by the legend in Figure 1. Please clarify/correct.

"A p-value of ≤0.001 was reported for the chi-squared test for the association between the calendar year and CrAg results" - does this imply that statistical significance was found in the trends for both a) CrAg volumes and b) CrAg positivity?

There doesn't seem to be statistical significance determination for the CrAg volumes during the discrete waves, nor for the CD4 median/categories, even though the conclusion in the abstract notes "A smaller impact on the median CD4 was noted" - what was the magnitude of the impact, and was it significant? The conclusion in the body of the manuscript seems to allude to significance by saying "...lower median CD4 counts during the respective COVID-19 waves". From a birds-eye (non-statistical perspective), the lack of change observed in distribution of CD4 categories would argue against a significant change.

The is a lot of data presented about the provincial percentage changes, however, the narrative is quite sparse. Did you have a hypothesis prior to analysing this data, and how did the findings relate to the hypothesis?

It may be useful in figures 2 and 3 to indicate the COVID-19 waves or lockdown periods (e.g. shading of relevant months).

The conclusion mentions "This confirms that patient CrAg testing overall was more affected by COVID-19 than CD4 testing." It is acknowledged in the earlier paragraph of the manuscript that the CD4 test volume data is not shown, but I think either the authors should show/reference the data, or remove mention of this.

"Another limitation of this study was that no linking of the CrAg and COVID-19 results" - would this really have been a valuable analysis to undertake given the differing incubation periods of COVID-19 and cryptococcosis?

I feel like there are more limitations than acknowledged:

- Was the reflex testing cohort chosen for convenience purposes? Although the non-reflex CrAgs probably do not contribute to a large number of tests, the selection of reflex only could introduce bias.

- It is not clear why a lower limit of 15 years was chosen for the sample. Guidelines for CD4 +/- CrAg include all adolescents from age 10 years and above, as far as I am aware of. Is this also related to convenience?

- The discussion does not take into consideration that there was a change in testing strategy implemented in 2019/2020. Most notably with newer guidelines, 6-monthly CD4 testing is indicated if VL > 1000. So although surveillance CD4 from the "wellness pre-ART" cohort is no longer a consideration, we could expect an increase in CD4 testing in 2019/2020 (compared to 2018) due to more frequent testing of patients who fail to achieve virological suppression.

Regarding non-availability of data due to lack of permission, why is this the case? What steps were taken to gain permission to share the data as part of the proposed publication?

---

## [Author Response · Author response to Decision Letter 0]

26 Jul 2023

All reviewer changes have been made to the manuscript and a detailed response is attached.

---

## [Editor Report · Decision Letter 1]

3 Aug 2023

PONE-D-22-31762R1Using laboratory data to assess the impact of Coronavirus (COVID-19) on reflex cryptococcal antigenaemia (CrAg) testing in South AfricaPLOS ONE

Dear Dr. Cassim,

Thank you for re submitting your manuscript to PLOS ONE. After careful consideration, we feel that it has merit but does not fully meet PLOS ONE’s publication criteria as it currently stands. Therefore, we invite you to submit a revised version of the manuscript that addresses the points raised during the review process. 1. Please correct the language issues i.e. line 140 do you mean date or data2. Please explain how samples from the same patient were handled?3. Your table and figure headings are too long. Please ensure that they are short and succinct.4. When describing data using the monthly median values it is standard practice to include the IQR.5. You conclusion should be renamed discussion and the conclusion should be short and succinct also covering how this study impacts practice.6. Please do not repeat results in discussion. Rather highlight main finding and discuss them.7. Table 4 belongs to results8. Please also include the health department ethical approval/ permission in your submission. Please submit your revised manuscript by Sep 17 2023 11:59PM. If you will need more time than this to complete your revisions, please reply to this message or contact the journal office at plosone@plos.org. Please include the following items when submitting your revised manuscript:A rebuttal letter that responds to each point raised by the academic editor and reviewer(s). You should upload this letter as a separate file labeled 'Response to Reviewers'.A marked-up copy of your manuscript that highlights changes made to the original version. You should upload this as a separate file labeled 'Revised Manuscript with Track Changes'.An unmarked version of your revised paper without tracked changes. You should upload this as a separate file labeled 'Manuscript'.If applicable, we recommend that you deposit your laboratory protocols in protocols.io to enhance the reproducibility of your results. Protocols.io assigns your protocol its own identifier (DOI) so that it can be cited independently in the future. For instructions see: https://journals.plos.org/plosone/s/submission-guidelines#loc-laboratory-protocols. Additionally, PLOS ONE offers an option for publishing peer-reviewed Lab Protocol articles, which describe protocols hosted on protocols.io. Read more information on sharing protocols at https://plos.org/protocols?utm_medium=editorial-email&utm_source=authorletters&utm_campaign=protocols.

We look forward to receiving your revised manuscript.

Kind regards,

Mergan Naidoo, PhD

Academic Editor

PLOS ONE
---

## [Author Response · Author response to Decision Letter 1]

29 Aug 2023

A detailed response has been attached.

---

## [Editor Report · Decision Letter 2]

30 Aug 2023

PONE-D-22-31762R2Using laboratory data to assess the impact of Coronavirus (COVID-19) on reflex cryptococcal antigenaemia (CrAg) testing in South AfricaPLOS ONE

Dear Dr. Cassim, Thank you for submitting your revised manuscript to PLOS ONE. After careful consideration, we feel that it has merit but does not fully meet PLOS ONE’s publication criteria as it currently stands. Therefore, we invite you to submit a revised version of the manuscript that addresses the points raised during the review process.1. Table 1 and 2 headings needs to be shortened2. Provide all IQRs when submitting median values in the manuscript. Please read through the manuscript and correct this e.g. page 9.

We look forward to receiving your revised manuscript.

Kind regards,

Mergan Naidoo, PhD

Academic Editor

PLOS ONE
---

## [Author Response · Author response to Decision Letter 2]

4 Sep 2023

A detailed response has been attached.

---

## [Editor Report · Decision Letter 3]

12 Sep 2023

Using laboratory data to assess the impact of Coronavirus (COVID-19) on reflex cryptococcal antigenaemia (CrAg) testing in South Africa

PONE-D-22-31762R3

Dear Dr. Naseem Cassim

We’re pleased to inform you that your manuscript has been judged scientifically suitable for publication and will be formally accepted for publication once it meets all outstanding technical requirements.

Kind regards,

Mergan Naidoo, PhD

Academic Editor

PLOS ONE
---

## [Editor Report · Acceptance letter]

19 Sep 2023

PONE-D-22-31762R3 

Using laboratory data to assess the impact of Coronavirus (COVID-19) on reflex cryptococcal antigenaemia (CrAg) testing in South Africa 

Dear Dr. Cassim:

I'm pleased to inform you that your manuscript has been deemed suitable for publication in PLOS ONE. Congratulations! Your manuscript is now with our production department. 

Kind regards, 

on behalf of

Professor Mergan Naidoo 

Academic Editor

PLOS ONE